# Qualitative Nitrogen Malnutrition Damages Gut and Alters Microbiome in Adult Mice. A Preliminary Histopathological Study

**DOI:** 10.3390/nu13041089

**Published:** 2021-03-26

**Authors:** Giovanni Corsetti, Claudia Romano, Evasio Pasini, Cristian Testa, Francesco S. Dioguardi

**Affiliations:** 1Division of Human Anatomy and Physiopathology, Department of Clinical and Experimental Sciences, University of Brescia, 25023 Brescia, Italy; cla300482@gmail.com; 2Cardiac Rehabilitation Division, Scientific Clinical Institutes Maugeri, IRCCS-Lumezzane, 25065 Lumezzane (Brescia), Italy; evpasini@gmail.com; 3Functional Point, Clinical and Virology Laboratory, 25121 Bergamo, Italy; krsnaj@gmail.com; 4Department of Internal Medicine, University of Cagliari, 9128 Cagliari, Italy; fsdioguardi@gmail.com

**Keywords:** malnutrition, gut, microbiota, amino acids, proteins, mice

## Abstract

Amino-acids (AAs) are the exclusive source of nitrogen for cells. AAs result from the breakdown of food proteins and are absorbed by mucosa of the small intestine that act as a barrier to harmful materials. The quality of food proteins may differ, since it reflects content in Essential-AAs (EAAs) and digestibility but, until now, attention was paid mainly to the interaction between indigested proteins as a whole and microbiota. The link between microbiome and quality of proteins has been poorly studied, although these metabolic interactions are becoming more significant in different illnesses. We studied the effects of a special diet containing unbalanced EAAs/Non-EAAs ratio, providing excess of Non-EAAs, on the histopathology of gut epithelium and on the microbiome in adult mice, as model of qualitative malnutrition. Excess in Non-EAAs have unfavorable quick effect on body weight, gut cells, and microbiome, promoting weakening of the intestinal barrier. Re-feeding these animals with standard diet partially reversed the body alterations. The results prove that an unbalanced EAAs/Non-EAAs ratio is primarily responsible for microbiome modifications, not vice-versa. Therefore, treating microbiota independently by treating co-existing qualitative malnutrition does not make sense. This study also provides a reproducible model of sarcopenia-wasting cachexia like the human protein malnutrition.

## 1. Introduction

Malnutrition results from eating an imbalanced diet that does not supply a healthy amount of one or more nutrients. Among nutrients, proteins play an irreplaceable role. Proteins are macromolecules made up of specific amino acids (AAs) sequences, serving a vast array of functions within the cell. Quantitative and qualitative reduced proteins intake, or an imbalance in protein synthesis and degradation, have been associated with severe depletion of body protein reserve, eventually resulting in malnutrition.

The AAs are fundamental for life, because they are the precursors for protein synthesis, since they are main source of nitrogen for mammals’ metabolism. Unfortunately, about half of the AAs in proteins cannot be synthesized or stored in metazoans, because the genes for their synthesis were lost early in evolution. These AAs are called essential (EAAs), and they must be taken exclusively through food. The other AAs can be synthesized in an autonomous way by the body, and therefore are defined non-essential (NEAAs). To best survive, all organisms must maintain a full complement of the AAs, best if with excess of EAAs [1,2,3].

AAs, resulting from the breakdown of dietary proteins, are absorbed almost completely in the initial tract of the small intestine, crossing the intestinal mucosa and reaching the capillary circulation of the villi. The intestinal mucosa is composed of a mucin layer covering the cells, enterocytes, and the apical junctional complex in between the cells [4]. Numerous experimental pieces of evidence suggest that alteration of barrier function is a potential pathway for intestinal and extra intestinal inflammation [5]. Indeed, intestinal mucosa acts not only in digestion and absorption, but also plays an important role as a barrier to toxic and harmful materials and protects from different antigenic and inflammatory reactions. Changes of intestinal barrier function were mainly due to the relaxation of the tight-junction between intestinal epithelial cells [6]. Different factors causing alterations in gut microbiota, hormones secreted by the enterocytes, and related changes of enzyme synthesis would all damage the intestinal barrier, with the enteric bacteria and endotoxin reinforcing those damages [7]. Disruption of the gut barrier in several physiological or pathological conditions would lead to altered exchanges of intraluminal solutes into the systemic circulation thus drives the organism to chronic diseases [8,9]. Indeed, gut microbiota alterations are also associated either with obesity [10], cancer [11], or with systemic inflammation, or even to etiology and pathogenesis of Kawasaki disease in children [12]. Microbiota modifications through age have been documented, and correlation among different grades of chronic disease and microbiome alterations have also been reported [13,14]. Furthermore, a recent pilot study showed that also patients with COVID-19 had persistent alterations in the fecal microbiome during hospitalization associated with severity of disease [15]. In addition, cumulative evidence shows a linkage between gut microbiota pattern and depression [16].

Although there is a growing interest and perception about mammalian microbiome and host relationship, most papers have focused mainly on fat and carbohydrate based nutrients, because partially digested proteins may reach, feed, and influence the different components of the microbiome [10,11,17]. Some attention was paid to the interaction between indigested proteins and microbiota [18], and, above all, to qualitative protein malnutrition. Indeed, qualitative protein malnutrition is not a rare event if we consider, in addition to chronic diseases as aging, cancer, etc., the spread of “junk foods” that invade the world market, or the “do it by yourself” diets, or diets that exclude or are significantly deficient in some macronutrients.

Microbiome and proteins’ biological quality should be peculiarly linked, and this link has been poorly studied, but some studies show a role for peculiar toxic substances derived by metabolic interactions of the microbiome with incomplete digestion products of food proteins, whose roles are becoming more and more significant in different illnesses [18,19,20]. Indeed, different amino acid compositions and digestibility of proteins, which are influenced by source, kinds of thermal processing, and amount of intake, should play a pivotal role in determining microbiota quality and quantities [21]. Therefore, we studied the effects of a special diet containing as unique nitrogen source a peculiarly balanced mixture of free essential amino acids (EAAs) and non-EAA (NEAAs), further expanding the excess NEAAs usually provided by food proteins, and compared the effects of this perfectly isocaloric diet to laboratory pellet and to control animals fed normal proteins, as a model of qualitative malnutrition. The aim was to determine whether qualitative and not quantitative malnutrition, that is, by diets providing the same nitrogen intake, but with excess in NEAAs and reduced EAAs, could induce alteration of gut epithelium morphology and enzyme as lysozyme, or cytokines as TNFα, TGFβ1, and NF-κB. As markers of altered permeability of gut wall, we measured the levels of zonulin in the epithelium and feces. In addition, the main kinds of microbiome (*Lactobacillus* spp., *Escherichia coli*, fecal coliforms, *Mycétes*, and *Clostridium* spp.) were evaluated. Finally, we observed whether re-feeding of qualitatively malnourished mice with standard diet could reverse the morphological, enzymatic, and microbiota alterations.

## 2. Materials and Methods

The experimental protocol was approved and conducted in accordance with the Italian Ministry of Health and complied with the ‘The National Animal Protection Guidelines’. The Ethical Committee for animal experiments of the University of Brescia and the Italian Ministry of Health had approved the procedures.

### 2.1. Diets and Mice

We designed a pellet containing all macronutrients in the same ratios, so that carbohydrates and lipids would have been similar and continuously provided to gut flora in the same ratios, to disentangle the relationships between caloric supply and macronutrients. Totally different was the quality of nitrogen intake containing as exclusive source of nitrogen a mixture of free EAAs (15%) and free NEAAs (85%) peculiarly enriched of arginine and glutamine. This diet was named Deficient Diet (DD) and produced in accordance with AIN76-A/NIH-7 rule (Dottori Piccioni, Milan, Italy). The DD was compared with the standard laboratory pellet (StD) (Mucedola s.r.l., Milan, Italy). The composition of the diets is resumed in Table 1. Twenty-eight six-month-old adult male Balb/C mice were randomly assigned to two groups. The first group (*n* = 8) was fed with StD ad libitum. The second group (*n* = 20) was fed with DD ad libitum. The animals were placed in a quiet, temperature- and humidity-controlled room and were kept on a 12/12-h light/dark cycle (lights on from 7 a.m. to 7 p.m.), and were daily inspected by skilled veterinarians. Every three days, the body weight (bw), food, and water consumption were measured. After 21 days, bw of DD-fed animals decreased by 30%, although daily caloric introduction was comparable to StD-fed animals. These animals were not suffering according to veterinary advice. Ten DD-fed animals were euthanized by cervical dislocation. The remaining ten DD-fed mice were re-fed with StD (DD-ref group) and after 30 days all animals was euthanized.

### 2.2. Samples Collections

In all groups, the blood glycaemia was measured at the end of each treatment. The samples of blood from heart, urine, and feces were collected immediately after euthanasia. The jejunum was immediately removed, quickly washed in physiological solution, immersed in immunofix, and embedded in paraffin for histochemistry (HC) and immune-HC (IHC) analysis.

### 2.3. Histology

Morphology is a good indicator of the status of the intestine [22]. For histopathological analysis, the gut was sectioned at 5 µm, slide mounted, and stained with H&E. The height of twelve villi and depth of twelve crypts were determined in each slide using the Olympus BX50 microscope equipped with image analysis system. According to [23], in gut we measured thickness of wall, tunica mucosa, tunica *muscularis*, and depth of intestinal crypts. Villi were measured linearly at the center of the villus from the basis at the crypt-villus junction to the villus apex, when there was a complete longitudinal section of a villus. The following measurements were obtained: (i) villus height, from the top of the villus to the villus-crypt junction; (ii) mucosal thickness, from the top of the villus to the border over the *muscularis mucosae*; and (iii) crypt depth, as difference between the total mucosal thickness and the villus height [24,25]. The number of goblet cells each mm of intestinal mucosa was determined by PAS staining [26,27]. The severity of inflammatory cell infiltrate was evaluated with a score from 1 (minimal, <10%) to 4 (marked, >51%), as previously described [28].

### 2.4. Immunohistochemistry

For IHC, gut sections were incubated overnight with primary anti-lysozyme (3349-1) from Epitomics Inc. (Burlingame, CA, USA), anti-Zonulin (ab191143) from Abcam (Cambridge, UK), anti-TNFα (NB600-587) from Novus Biologicals (Centennial, CO, USA), anti-TGFβ1 (sc-146) Santa Cruz Biotechnology Inc. (Dallas, TX, USA), anti-NF-κB (NB110-57266) from Novus Biologicals; all polyclonal antibodies were diluted 1:100 with PBS; monoclonal antibodies were diluted 1:250, with PBS. The sections were processed according to the manufacturer’s protocol and visualized with a rabbit ABC-peroxidase staining system kit (Santa Cruz Biotechnology Inc., Dallas, TX, USA). In order to exclude incorrect interpretation of immunostaining due to endogenous biotin, we also carried out experiments using the peroxidase-anti-peroxidase detection system. We obtained similar results with both methods. Each set of experiments was performed in triplicate, with each replicate carried out under the same experimental conditions. The IHC control was performed by omitting the primary antibody in the presence of isotype-matched IgGs. The staining intensity in both HC and IHC slides was evaluated using an optical Olympus BX50 microscope equipped with an image analysis program (Image Pro Plus 4.5.1, Immagini e Computer, Milano, Italy) and analyzed quantitatively. The IOD was calculated for arbitrary areas, by measuring 10 fields for each sample using a 20× lens. On sections stained with anti-NF-κB, we counted the percentage of immunostained enterocytes nuclei analyzing over 30 fields for each group using a 20× lens. Intestinal permeability was evaluated as a fecal zonulin concentration (ng/mL) using commercial ELISA kit (Zonulin, Stool, ELISA, DRG Instruments GmbH, Marburg, Germany).

### 2.5. Microbiota

Gut flora was determined by the development of bacteria (*Lactobacillus* spp., *Escherichia coli*, fecal coliforms, and *Clostridium* spp.) and *Mycétes* in stools, as previously described [29]. Stool samples were collected from intestine after euthanasia with strikers, and inserted into hermetic vials using a specific medium. Subsequently, the microbiota was measured after 48 h of incubation under proper conditions using a selective agar. Further proof of isolation was performed by using bacterial metabolic tests on isolated organisms through the BBL Crystal Identification System (Becton Dickinson, Franklin Lakes, NJ, USA). The results are expressed in colony-forming units (c.f.u.) per milliliter of stool. Test was performed by Functional Point (Bergamo, Italy), a clinical and virology laboratory that adheres to international quality control standards and is accredited as an official laboratory within the National Health System. The test coefficient of variation was <9%.

### 2.6. Blood and Urine Analysis

These analyses were performed by “Division of Laboratory Animals” of Istituto Zooprofilattico Sperimentale della Lombardia e dell’Emilia Romagna di Brescia (IZSLER-Bs). Blood and urine samples were collected immediately after euthanasia in all groups. Blood samples were collected into tubes containing EDTA anticoagulant and without anticoagulant. Hemochromocytometric value was measured by Cell-Dyn 3700 laser-impedance cell counter (Abbott Diagnostics Division, Abbott Laboratories, Chicago, IL, USA). From blood serum and urine, urea, albumin, and creatinine concentration were measured by a biochemical automatic analyzer ILab Aries (Instrumentation Laboratory, Lexington, MA, USA) and its ready-to-use kits. The ratio between neutrophils (N) and lymphocytes (L) has been used as inflammation index.

### 2.7. Statistics

Data are expressed as mean ± SD. Statistical analysis was performed by one-way ANOVA followed by the Bonferroni test or by a Student *t*-test, to compare the results of the different experimental groups. A value of *p* < 0.05 was considered statistically significant.

## 3. Results

### 3.1. Phenotypical Modifications and Food Consumption

#### 3.1.1. DD Fed Animals

Mice fed with DD lost 30% of bw after three weeks (Figure 1A), and then were regularly and most frequently evaluated by veterinarians controlling health status, if they were not suffering and maintaining normal daily and nocturnal activities. No differences in food (g/day) consumption were seen between StD and DD. However, from 14° to 21° days, in front of linear progression of weight loss, DD-fed animals progressively showed an eating behavior of a more elevated amount of food and caloric intake than control animals (Figure 1B). Nevertheless, in the DD group, there was a progressive and significant decrease of stool weight (Table 2), which was carefully investigated if eventually correlated to abnormal food ingestion by controlling even minimal presence of fragments of wasted pellet while cleaning cages. Mean body weight, organs, and muscle (*triceps surae*) weight decreased significantly compared to StD-fed mice; those values are summarized in Table 3.

#### 3.1.2. DD-Ref Animals

Those mice recover weight approaching weight of control mice in about 10 days; still, it remains consistently lower than in control ones (Figure 1A). Stool production increases, but remains lower than StD-fed mice (Table 2). The consumption of food increases strongly (about +70%) the first day after the switch to normal refeeding diet, but decreases in subsequent days (Figure 1B). Organ weights from DD-ref mice increased, but remained significantly lower than those of control animals. However, WAT and BAT increased exceeding that of control animals (Table 3).

### 3.2. Blood Analysis

Comparison of blood and serum data between groups are summarized in Table 4. With exception of hemoglobin, all other blood parameters changed significantly in DD mice. In particular, we observed a mild increase in neutrophils (N) concentration and marked decrease in lymphocytes (L) concentration. Therefore, the ratio N/L, an inflammation index, was higher than in StD-fed mice. In serum, the albumin and urea concentration did not change significantly. Creatinine increased significantly, whereas aptoglobin strongly decreased. After refeeding, almost all blood data reached values not different to control ones.

### 3.3. Urine Analysis

Comparison of urine analyses between groups are summarized in Table 5. In DD-fed mice, we observed the decrease of filtered albumin, creatinine, and total proteins, whereas urea excretion increased significantly. After refeeding, both albumin and total protein concentrations returned to standard values. On the contrary, urea and creatinine values remained still higher than in StD-fed mice.

### 3.4. Fecal Microbiome

The predominant microbiota isolated from feces in StD-fed animals was *Lactobacillus* spp. (over 1,000,000 × 10^3^ c.f.u./g), representing 99.99% of the microbiota examined. Fecal coliforms, *E. coli* and *Mycètes* spp., were very poor (1000 × 10^3^ to 5000 × 10^3^ c.f.u./g) and globally represent 0.01% of the microbiota. The *Clostridium* spp. were absent. Conversely, the DD-fed animals showed altered relationship among components of microbiota. *Lactobacillus* spp. decreased about 5%. On the contrary, fecal coliforms increased about 134 fold (0.67% of total), *E. coli* about 185% (0.74% of total), and peculiarly, *Mycètes* spp. increased about 312 fold (1.56% of total). Re-feeding induces rapid inhibition of fecal coliforms, *E. coli* and *Mycètes* spp., already marked after seven days, coupled to increased numbers of *Lactobacillus* spp. After 30 days, DD-ref mice regained a microbiota composition comparable to those of starting (Figure 2A–D).

### 3.5. Morphology and Histopathology of the Jejunum

Morphometric data and variables of the gut wall according to groups are resumed in Table 6.

#### 3.5.1. DD-Fed Animals

Marked, but not significant, tendency to decrease in total intestinal wall thickness was observed in DD-fed mice. In addition, also no differences were observed between the groups in the mucosa layer thickness. However, in DD-fed mice, *tunica muscular* is thinning, and almost all myocytes nuclei that appeared stocky and “caterpillar-like” shape were instead that elongated and “earthworm-like” shape as in StD-fed mice (Figure 3A,B). The villi length, diameter, and crypt depth did not change. However, the epithelium was thinner than in StD-fed mice (Figure 4A,B). The nuclear diameter of enterocytes showed the tendency to increase, whereas the number of nuclei (nuclear density) decreased significantly (Table 6), as well as the junctions between enterocytes (Figure 5A,B,D). Number of PAS-positive goblet cells (nr/mm of epithelial layer), significantly decrease in DD-fed gut than in the StD-fed gut (Figure 6A,B,D). Furthermore, DD-fed animals showed Paneth cells with containing few small granules (Figure 7A,B,D). In these animals, the mucosa showed moderate (score 3) amount of inflammatory cells infiltrate. Some villi showed marked inflammatory infiltrate (score 4) (Figure 8A,B).

#### 3.5.2. DD-Ref Animals

Mice refeeding with StD improves the parameters concerning the thickness of muscular layer, the nuclear shape of myocytes (Figure 3C), epithelium thickness, nuclear diameter of enterocytes (Figure 4C and Table 6), goblet cells (Figure 6C,D), and granules of Paneth cells (Figure 7C,D). However, the number of enterocytes nuclei and their junctions (Table 6 and Figure 5C,D) still remained lower than in StD. Occasionally, minimal (score 1) to mild (score 2) inflammatory infiltrate is still present (Figure 7C).

### 3.6. Immunohistochemistry

Lysozyme immunostaining was intense in all Paneth cells at the bottom of the crypts of Lieberkühn in the StD-fed animals. In contrast, the staining decreased significantly in DD-fed mice where only limited immunostained areas were observed. Refeeding increased lysozyme expression that in some cells appears to be markedly improved (Figure 9A–D).

In StD-fed mice, zonulin was evident in apical side of enterocytes (Figure 10A), while a tendency to decrease was observed in DD-fed animals where cytoplasm staining was faint (Figure 10B). After refeeding, zonulin immunostaining was more intense in the apical and basal side of enterocytes (Figure 10C). Evaluation of optical density of immunostaining according to groups is shown in figure (Figure 10D). On the contrary, fecal zonulin concentrations increased significantly, about three times in DD-fed mice. After refeeding, the fecal zonulin concentrations decreased, although remained significantly higher when compared to StD-fed mice (Figure 10E).

Staining for TNFα shows a notable increase in DD-fed animals when compared to StD-fed animals (Figure 11A,B). The staining intensity decreased with refeeding (Figure 11C). The optical density of immunostaining according to groups is shown in figure (Figure 11D).

TGF-β1 staining is diffusely and moderately intense in StD-fed animals (Figure 12A). Qualitative malnutrition induces the decrease of staining intensity, and the cytoplasm of enterocytes shows very faint staining (Figure 12B). Refeeding restores the intensity of immunostaining, which was significantly more elevated than in StD-fed animals (Figure 10C). The optical density of immunostaining according to groups is shown in Figure 12D.

In StD-fed mice, NF-κB staining was faint inside cytoplasm, and only some nuclei were moderately stained (Figure 13A,D). Of note, NF-κB staining inside cytoplasm was increased, but principally numerous nuclei of enterocytes from DD-fed mice were intensely stained (Figure 13B,D). Cytoplasm and nuclear staining returns to baseline levels by refeeding (Figure 13C,D).

## 4. Discussion

In this study, we have demonstrated that diets providing the same macro and micro nutrients quantities and the same nitrogen content as normal diets, but providing altered EAAs to NEAAs ratios (EAA/NEAA), through excess of NEAAs induce alterations of intestinal epithelium with concomitant altered microbiota relationships. Simultaneously, we found an experimental model of qualitative protein malnutrition capable of inducing quickly sarcopenia-wasting-cachexia syndrome, paralleled by a rapid modification of gut microbiota.

A first result of the present study is that DD-fed mice, although fed by an isocaloric diet and providing the same amount of nitrogen as StD, resulted in rapid body weight reduction, despite the consumption of food being comparable to that of StD-fed animals. These data confirm what emerged in previous studies, where it was shown that the diet rich in NEAAs and deficient EAAs is fully appreciated by animals [3], and are in contrast with other studies that argued that animals receiving an EAA-deficient or imbalanced diet fail to grow because they they would refuse the diet [30]. We showed that body weight loss was not due to rejection of diet but, most likely, to the need of qualitatively malnourished mice to recover EAAs and energy production by destroying endogenous proteins, especially muscle ones, as demonstrated by the significant reduction of muscle mass and adipose tissue with increase of serum creatinine concentration.

Adequate nutrition is necessary to maintain the gastrointestinal barrier, because this favors the enterocytes turnover, improving the normal cell division and migration from the crypt to the villi. It is well known that protein malnutrition is associated with villous atrophy, abnormal mucin formation, and impairment in the secretion of IgA during development, adulthood, and ageing [31,32]. Numerous studies have focused on quantitative protein malnutrition, finding that it disrupts the normal ecology of the microbiota, impairs host immune response and antibacterial defenses, enhances susceptibility to infection, and leads to mucosal atrophy, altering the gut barrier function and inducing bacterial translocation [33,34]. It was demonstrated that diet with reduced content of protein (75%) and vitamins (50%) given to weanling Wistar/NIN male rats for 20 weeks increased intestinal cell apoptosis [35]. Furthermore, maternal malnutrition altered expression of genes that maintain maternal gut homeostasis, and altered fetal gut permeability, function, and development [36]. Cancer cachexia is also reported to induce alterations of morphology in small intestine [37]. In addition, many studies were performed on the intestinal barrier using a number of techniques to evaluate eventually altered function [38], and a broad range of individual food substances have been tested in Caco-2 monolayers with divergent effects readings the trans epithelial electrical resistance and epithelial flux parameters [39]. However, these studies and many others, unlike the present study, represent models of quantitative, not qualitative, malnutrition.

Diet containing NEAA in excess quickly change the morphology of epithelium’s cells: the enterocytes, goblet, and Paneth cells. Enterocytes are columnar absorptive cells and have an apical striated border of microvilli. The enterocytes own the apical junctional complex that consists of a network of tight junction proteins and the adherent’s junction [40] that were anchored in the cell by means of filamentous actin cytoskeleton [41]. Tight junctions are pivotal elements that govern trans-epithelial transports. These junctions are very dynamic structures, and their dysregulation is involved in development of pathological condition as autoimmune diseases. Intracellular tight junction proteins are called zonulin proteins because they form the *zonula occludens*. Zonulin proteins link the cell cytoskeleton to the transmembrane tight junction proteins such as claudins, occludin, and junctional adhesion molecules that are mainly responsible for the intestinal barrier function [42].

Zonulin was identified as endogenous modulator of tight junction’s integrity in the small intestine, consequently regulating the permeability of the intestinal epithelium, favoring the movement of fluid, macromolecules, microorganisms, and leucocytes between the bloodstream and the intestinal lumen and vice versa. Therefore, dysregulation of zonulin synthesis may contribute to weakening tight junctions and favor disease states as autoimmune diseases, inflammation, and malignant transformation [43,44].

Changes in zonulin levels correlated with in vivo increased intestinal permeability [45]. Reduced expression of zonulin was reported in ulcerative colitis and Crohn’s disease patients, where the pentalaminar structure of tight junction was destroyed [46]. Zonulin is also altered by toxic substances. Immortalized intestinal cell lines (Caco-2) treated with ethanol (from 0.1 to 10%) induced progressive disruption of tight junction with increased elimination of zonulin by fecal route, and formation of large gaps between the adjacent cells [47]. Recent data show that zonulin-dependent small intestinal barrier impairment is an early step potentially linked to development of chronic inflammatory diseases [48]. In light of above evidences, our data show that a diet providing NEAAs in increased ratio of total nitrogen intake, thus lowering EAAs intake, reduces both intracytoplasmic zonulin and increases fecal concentration of zonulin, suggesting the possible alteration of intestinal permeability and a compromised intestinal barrier, possibly correlated to inflammation.

Potentially related to reduced enterocytes zonulin immunoreactivity, we also documented increased inflammatory reaction by TNF-α staining and massive nuclear translocation of NF-κB, and both are relevant findings, since TNF-α exerts a direct effect on gut barrier integrity. Indeed, mice fed with high-fat diet increased TNF-α mRNA expression in the intestine [49], while patients with Crohn’s disease receiving anti-TNF-α treatment improves gut barrier function [50]. Furthermore, TNF-α reduced trans-epithelial resistance and Zonulin-1 expression via a NF-κB-dependent pathway in Caco-2 cells [51]. Additionally, increased activation of NF-κB is detected in cells of tissues affected by chronic inflammation, where it is believed to exert detrimental functions by inducing the expression of pro-inflammatory cytokines that regulate and sustain the inflammatory response and could cause tissue damage [52]. Intestinal microbiota is believed to regulate the level of NF-κB activity at the epithelial interface and thereby affects the mucosal immune balance [53,54]. Effect of TNF-α on NF-κB activation also increased the expression and activity of myosin light chain kinase, which leads to disorganization of tight-junction proteins, thus impairing the intestinal barrier [55]. The NF-κB pathway, which is activated by alteration of intestinal microbiota, plays an important role in activating host pro-inflammatory responses [56], and the role of inflammatory pathways appears to be critical in the regulation of gut barrier function [57], leading to leaky gut syndrome. The harmful effects of pro-inflammatory cytokines on the intestinal barrier can be ameliorated by antibiotic administration, suggesting either gut microbiota changes as causative and some correlation among the innate immune system, gut inflammation, and microbiota profile [57].

Finally, in animals fed with NEAAs-rich diet, we also observed the depletion of TGF-β1 staining. TGF-β1 is a cytokine synthesized and released by cells with efficient intestinal epithelial barrier function [58]. Significance of TGF-β1 expression had been previously described in wound healing [59]. In a healthy gut, a balance between regulatory and inflammation function would be necessary to maintain gut barrier integrity. Within the gut mucosa, TGF-β1 plays a pivotal role in mediating these balanced responses, since it regulates the function of many mucosal cell types in either autocrine and paracrine manner [60,61]. Of interest, TGF-β1 promotes the expression of tight junction protein and adhesion molecules, thus improving epithelial barrier integrity: inhibition of TGF-β1 favors gut inflammation [62]. In addition, it has been proven that mice with selective deletion of TGFβ1 signaling in gut epithelium are more susceptible either to colitis induced by dextran sodium sulphate or to invasive intestinal tumors [63]. TGF-β levels in a mammal’s gut are directly and indirectly modulated by the microbiota and microbiota-derived products with impacts on the development and functions of immune cell subsets, which in turn regulates microbiota sequestration within the gut lumen [64]. As predictable by literature, we observed the tendency to decrease in TGF-β1 immunostaining following qualitative protein malnutrition, and a concomitant decrease of zonulin and increased inflammatory state. Therefore, excess in NEAAs, or most probably, insufficient availability of EAAs through diet can severely impair the gut barrier, and so malnutrition could pave the way for serious diseases. Marked rise in TGF-β1 staining observed after refeeding suggest restoration of epithelial barrier integrity consequent to rebalancing EAAs to NEAAs ratios.

Previously, in mice models of intestinal obstruction, pre-treatment with a diet containing arginine (2%) preserved intestinal barrier integrity and reduced bacterial translocation in gut wall [65]. Additionally, glutamine has been indicated as a possible adjuvant therapy in different gut illnesses [66]. Our NEAAs-rich diet contained high amounts of arginine (11.8%), and also of glutamine (10.2%); nevertheless, although we have not measured bacterial translocation, we observed a significant alteration of the intestinal mucosa by the loss of zonulin with the stool, suggesting increased permeability of the barrier. An increased supply of arginine and of glutamine, therefore, was not sufficient to protect intestinal barrier integrity when EAAs provided by diets are low. We believe that this discrepancy is due to the fact that our diet contained only 15% EAAs (EAA/NEAA << 0.2), and no study dealing with supplementation of glutamine or arginine has tested this kind of supplementation against EAAs enriched diets (EAA/NEAA ≥ 1). Indeed, in experimental models, we have repeatedly observed that even slight reductions of concentrations of EAAs under the normal content of food proteins (near to 45% or EAA/NEAA < 0.9–0.7) induces proportionally significant alteration both of organ morphology and hematologic parameters [1,3].

Goblet cells secrete mucin, a glycoprotein that forms a protective layer of intestinal lumen. Indeed, mucin forms a mucus layer, which keep separate materials in cavities from the intestinal epithelium and acts as a lubricant. However, mucin also constitutes a physical barrier that traps and therefore prevents invasion of pathogenic microorganisms. In addition, goblet cells participate in immune response through nonspecific endocytosis and goblet cell-associated antigen transfers [67,68]. Altered mucin production and consequent dysfunction of the mucosal barrier are related to the occurrence of inflammatory diseases such as ulcerative colitis and Crohn’s disease [69,70]. We observed that diet providing excess of NEAAs, and so a reduced ratio of EAA/NEAA, decreased goblet cells’ number, hence the following thinning of the mucin layer rapidly exposing intestinal epithelium to a greater risk of inflammation and bacterial attack. This also may result in impairment of intestinal absorption. In effect, the depletion of goblet cells leads to increased adhesion of bacteria to the surface of the epithelium and so reduces digestion and absorption of nutrients, because it alters gut barrier efficiency [71,72]. Certainly, the decrease of lysozyme and secretory granules in Paneth cells observed in our mice fed with NEAAs-rich diet contributed to exacerbating any possible inflammatory response due to impairment of the gut barrier. In fact, normality was restored by returning to standard diet. Those findings are of interest, since current evidence suggests that defective Paneth cells may play a key role in promoting gut inflammation and also in Crohn’s disease, by allowing bacterial attachment and invasion [73].

Gut microbiota is the assembly of microorganisms living in our intestine. The correct composition and functionality of the microbiome is essential for maintaining a “healthy status”. The profile of the gut microbiota is influenced by numerous variables, including requirements of metabolism substrates, microbiological factors, and environmental factors. Based on our findings, however, the variable that may have the greatest impact on the composition of microbiota is adequacy of EAAs provided by diet. Dietary factors, and particularly matching nitrogen needs, can enhance gastrointestinal health by shaping the nature of gut microbiota [34,74]. The resident species of gut microbiota may use AAs derived from food proteins or from the host as elements for new protein synthesis; in addition, they drive nutrient metabolism by conversion or fermentation. Of note, gut microbiota can synthesize several nutritionally useful EAAs, and this is a potential regulatory factor in AAs homeostasis [74], but may produce also uremic toxins [18]. Indeed, part of the whole protein ingested with the diet is not digested, and peculiarly its content in tryptophan is transformed into toxic indolamine by indolamine 2,3-dioxygenase (IDO) via the kynurenine pathway. This suppresses T cells, activates Treg cells, and promotes inflammatory processes [75]. In contrast, the free tryptophan contained in the special mixture of EAAs is completely absorbed in the ileum and transported in the blood by albumin [76]. Keeping tryptophan plasma concentration constant inhibits IDO and consequently blunt endogenous uremic toxin syntheses and systemic inflammation [77].

In a sample of healthy individuals consuming unbalanced (quantitative) protein diet (high-protein and low-protein diet) for 42 days, reduction mostly in the pool of bacteria belonging to the *Bifidobacteria* genus [78,79] was reported. In another study, milk-fed piglets after weaning were fed a soybean based diet, and despite the good health status, owing to the buffering effect of the protein, those animals exhibited lowered gastrointestinal pH and a decreased *Lactobacillus*-to-*Coliforms* ratio [80,81]. It is interesting to note that soybean contains NEAAs in excess of EAAs, with the ratio EAA/NEAA being around 30/70 (or <0.43). This is in line with what was highlighted by our data, since diet rich in NEAAs, although providing high amounts of arginine and glutamine, determined an unfavorable ratio between *Lactobacillus* spp. and *Coliforms* spp., but the rise in number of *Mycétes* was by far more relevant. A more balanced refeeding, and peculiarly less reduced EAA/NEAA containing diets restore the predominance of *Lactobacillus* spp. over other species. Higher *Lactobacillus*-to-*Coliforms* ratio indicates a favorable proportion of advantageous *Lactobacillus* spp. relative to *Coliforms* spp., a population that could include coliform pathogens [81]. Indeed, it was proposed that *Lactobacillus* strains play an important role in development of anti-infectious agents that act luminally and intracellularly in the gastrointestinal tract [82].

*Mycétes* spp. reside permanently in the intestine and, in healthy subjects, act synergistically with other microbiota to maintain homeostasis. Correlation between alterations in gut *Mycétes* and pathogenesis of inflammatory bowel disease, such as Crohn’s disease and ulcerative colitis, has recently been reviewed [83,84]. In addition, also patients with chronic heart failure may have intestinal overgrowth of pathogenic bacteria and *Candida* spp. [85]. Furthermore, also diabetes type 2 may induce significant intestinal *Mycétes* overgrowth, increasing intestinal permeability and systemic low-grade inflammation [86]. In our study, *Mycétes* are modified as a consequence of low quality protein intake, but return to their initial state with refeeding with standard proteins. We believe that this finding is of particular importance, as it suggests that low quality protein diet promotes the development of fungi, and also that adequate supplementation of even a poor diet with EAAs (so increasing EAA/NEAA ratio) could be a way to rapidly reduce intestinal inflammatory status. Therefore, modulation of the microbiota and *Mycétes* can be considered as a therapeutic approach for treating gut inflammation and related diseases, even in patients with multiple pathologies. On the other hand, we propose that each strategy aiming to obtain the homeostasis of the microbiota should consider carefully modifications and content of *Mycétes*.

In summary, the microbiota and intestinal wall represent a peculiar morpho-functional unit, whose characteristics and role are partially unknown. At present, it is difficult to understand how relevant specific alterations of intestinal mucosa would be in determining an alteration of the microbial flora or vice versa. In any case, we provided evidence that damages induced by nutritional deficits, and chiefly by EAAs deficits, that is by a qualitative malnutrition, definitively harm both intestinal wall integrity and equilibrium of microbiota composition.

### Study Limitation

A possible limitation of this study is the exclusive application of IHC to support our results. However, as applied in a previous spleen study [2,87], this choice was dictated by the fact that the gut also contains heterogeneous cell population (enterocytes, goblet cells, Paneth’s cells, myocytes, etc.). Consequently, histopathological changes and especially the exact location of markers is possible alone with IHC and HC. In contrast, molecular analysis, although much more sensitive in highlighting the presence of proteins and very often used exclusively, does require immediate freezing and homogenization of the sample. As a consequence, it does not take into account the specificities of the proteins location such as the tissue morphology and organization. This is a major limitation in the exclusive use of molecular analysis, which we believe is comparable if not superior to IHC. For these reasons, we believe that our data, even if obtained only by IHC, are worth considering forming the basis for further studies.

## 5. Conclusions

To our knowledge this study is the first to examine the effect of qualitative and not only quantitative protein malnutrition. Qualitative malnutrition was induced by a nutritionally balanced diet significantly enriched in NEAAs. Results reveal that altered EAAs to NEAAs ratios (EAA/NEAA) have a quick and unfavorable impact on body and organ weight, gut cells and microbiota, and *Mycétes* spp*. in primis*, since they promote peculiar weakening of the intestinal barrier. Refeeding with diet providing a standard quality content of proteins, that is, increasing EAAs content, allows the recovery of gut integrity and a healthier balance in microbiota composition in a relatively short time; still, it does not completely restore body weight. This model is a proof of concept that nitrogen quality and related malnutrition are primarily responsible for microbiota modifications, and not vice-versa. Therefore, we propose that any attempt to treat microbiota independently by treating also co-existing qualitative malnutrition is unreasonable. This model also proves that a relatively short period of qualitative protein malnutrition, that is, diets providing poor quality protein content, may consistently alter the anatomy of fundamental organs of living beings, and peculiarly intestine integrity. Therefore, our special diet rich in NEAAs provide a rapid and reproducible experimental model of sarcopenia-wasting-cachexia, similar to the clinical picture of prolonged human protein malnutrition.

Finally, this work may suggest research on dietary interventions even in humans, for the purpose of preventing or mitigating the effects of non-optimal nutrition above all in patients affected by chronic gut inflammation and related diseases. Indeed, increased intestinal permeability can trigger gastrointestinal dysfunction and pathology, including recrudescence of Crohn’s disease, multiple organ dysfunction, bacterial translocation, food allergies, and acute pancreatitis [57,88,89].

The knowledge of interactions between microbiota and host metabolism, as well as understanding of modification of microbial ecology, would be certainly beneficial and provide effective therapeutic options for many diet-related diseases, and would further develop in the near future. Peculiar attention and further studies should specially focus on those vulnerable populations wherein qualitative malnutrition and infection are more likely to coexist, this exacerbating the noxious states.

## Figures and Tables

**Figure 1 nutrients-13-01089-f001:**
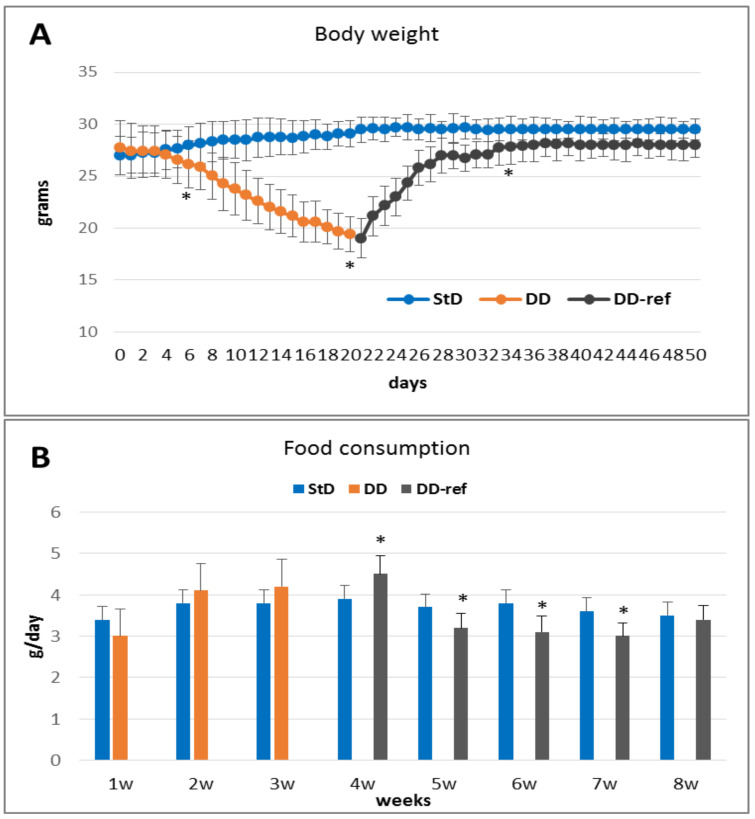
(**A**) Change in body weight (mean ± st.dev.) according to diet. After 21 days, DD-fed animals decreased their body weight by about 30% (orange line). Refeeding (grey line) quickly increased body weight; however, it remains lower that StD-fed mice (blue line). (**B**) Food consumption (g/day, mean ± st.dev.) according to diet. There are no differences between DD and StD consumption, although DD tended to be consumed more (orange columns). Curiously, during refeeding, StD it tended to be consumed less (grey columns) than control-fed mice (blue columns). * *p* < 0.05 vs. StD.

**Figure 2 nutrients-13-01089-f002:**
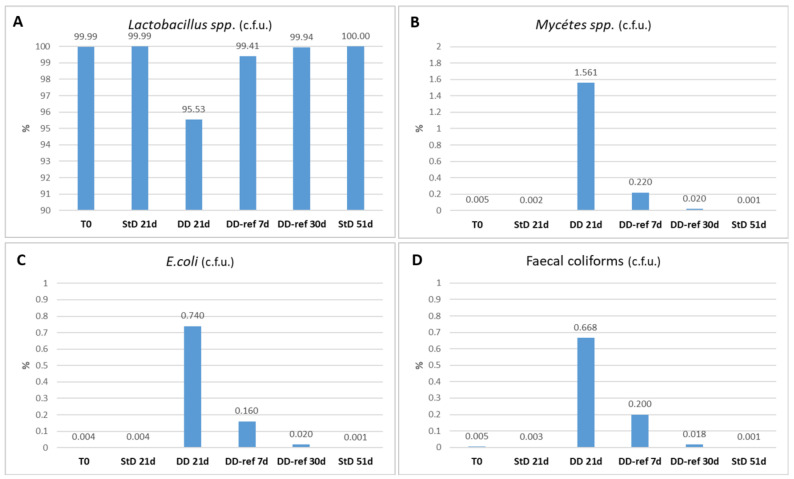
Microbiota relationship (% of c.f.u./g). DD-fed mice show a strong decrease in *Lactobacillus* spp., which are restored near the normal value after seven days of refeeding (**A**). *Mycètes* spp. (**B**) increases much more than *E. coli* (**C**) and coliforms (**D**) in DD-fed animals. Refeeding partially restored the microbiota concentration after seven days, reaching the standard value after 30 days.

**Figure 3 nutrients-13-01089-f003:**
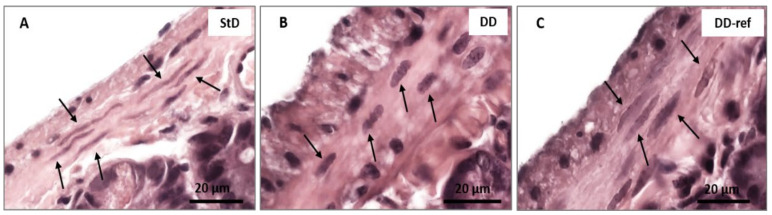
Myocytes nuclei of gut muscular wall (arrows) according to diet (H&E staining). (**A**) StD-fed mice show all nuclei with elongated and “earthworm-like” shape. (**B**) Stocky nuclei with “caterpillar-like” shape is characteristic of DD-fed mice. (**C**) Refeeding partially restores the nuclear shape. Scale bar = 20 µm.

**Figure 4 nutrients-13-01089-f004:**
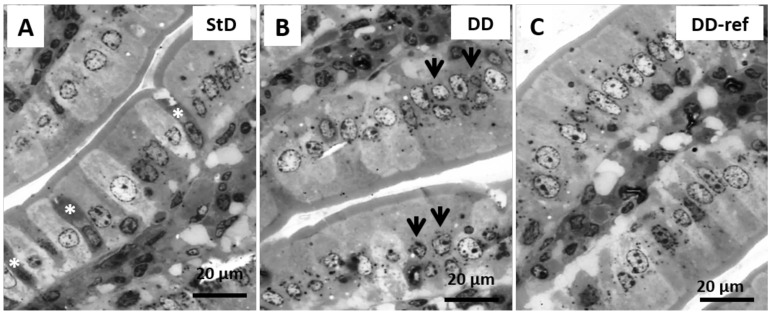
Representative images of epithelial layer from small intestine (semi-thin sections toluidine-blue stained, thickness 1 µm). StD-fed animals showed regularly organized absorptive epithelium and brush border. Enterocytes are well aligned, with regular size and nuclei of regular shape. The cytoplasm is devoid from waste inclusions. Paneth cells (asterisk) are regularly distributed (**A**). DD-fed mice showed evident epithelial disarrangement: the absorptive epithelium appears thinner with smaller enterocytes. Furthermore, enterocytes frequently present double and smallest nuclei (arrows). Many dark granules, similar to waste products, are present inside the cytoplasm (**B**). Refeeding restores the epithelium thickness, nuclear shape and size, and the regular and orderly distribution of enterocytes. Dark intracytosplamatic granules are still widely present (**C**) (original magnification 100× oil, scale bar 20 µm).

**Figure 5 nutrients-13-01089-f005:**
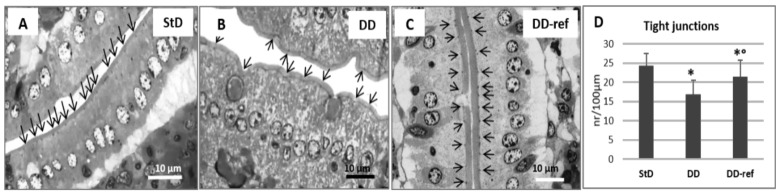
Tight junctions between enterocytes (arrows). Semi-thin sections toluidine-blue stained, (thickness 1 µm). (**A**) Tight junctions in StD-fed mice are easily identifiable by their higher color density (black dots) and appear regularly spaced. (**B**) After DD administration, the junction between enterocytes is less visible and therefore appears to be distributed unevenly. However, the brush border maintains a normal morphology. (**C**) Refeeding partially restores the intensity and regularity in the distribution of tight junctions. Scale bar = 10 µm. (**D**) Graph representing the number of thigh junctions each 100 µm of epithelial length according to diets. * *p* < 0.05 vs. StD, ° *p* < 0.05 vs. DD.

**Figure 6 nutrients-13-01089-f006:**
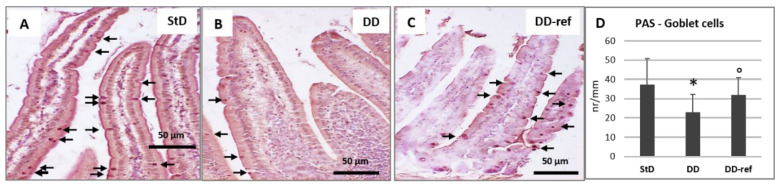
Goblet cells (PAS staining) (arrows). Goblet cells appear numerous in the StD (**A**), while in animals fed with DD, they significantly decrease (**B**). Their number increases moderately after refeeding (**C**). Scale bar = 50 µm. (**D**) Graph representing the number of Goblet cells each mm of epithelial length according to diets. * *p* < 0.05 vs. StD, ° *p* < 0.05 vs. DD.

**Figure 7 nutrients-13-01089-f007:**
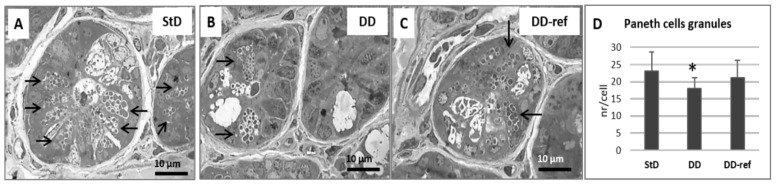
Paneth cells (Semi-thin sections toluidine-blue stained, thickness 1 µm). In StD-fed mice, Paneth cells appear engulfed by numerous secretory granules of different size (arrows) (**A**), whereas in animals fed with DD, the secretory granules tend to decrease (**B**). Moderate increases of secretory granules were seen after refeeding (**C**). Scale bar = 10 µm. (**D**) Graph representing the mean number of secretory granules in each Paneth cells according to diets. * *p* < 0.05 vs. StD.

**Figure 8 nutrients-13-01089-f008:**
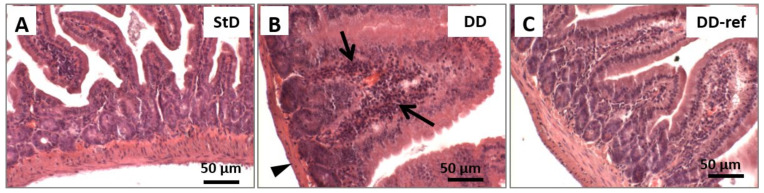
Representative images of H&E-stained of the small intestine sections illustrate the presence of inflammatory cell infiltrates. (**A**) StD-fed mice did not show inflammatory cell infiltration, and the morphology of mucosa appear normal. (**B**) In contrast, in animals fed with DD, the mucosa showed increased amount infiltration of mixed inflammatory cells (arrows). In addition, these animals showed thin muscular layer of gut wall (head arrow). (**C**) Refeeding restored the standard condition, although, occasionally, mild inflammatory infiltrate is still present (scale bar 50 µm).

**Figure 9 nutrients-13-01089-f009:**
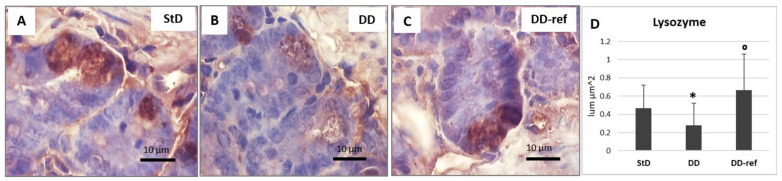
Lysozyme immunostaining. Intense staining is expressed by Paneth cells and secretory granules in the StD-fed mice (**A**). In contrast, DD-fed mice decreased staining significantly (**B**). Refeeding strongly increased immunostaining (**C**). Scale bar = 10 µm. (**D**) Graph representing the mean value of optical density according to diets. * *p* < 0.05 vs. StD, ° *p* < 0.05 vs. DD.

**Figure 10 nutrients-13-01089-f010:**
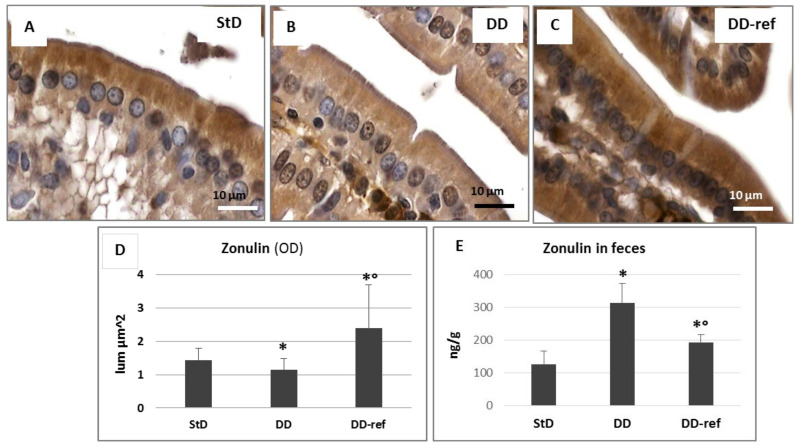
(**A**–**D**) Zonulin immunostaining is intense and mainly concentrated in the apical part of enterocytes (**A**). Feeding with DD significantly reduces the immunostaining intensity (**B**), whereas the strongly massively increases it (**C**). Graph representing the mean value of optical density according to diets (**D**). Scale bar = 10 µm. (**E**) graph representing the concentration of zonulin (ng/g) according to diet in feces. * *p* < 0.05 vs. StD, ° *p* < 0.05 vs. DD.

**Figure 11 nutrients-13-01089-f011:**
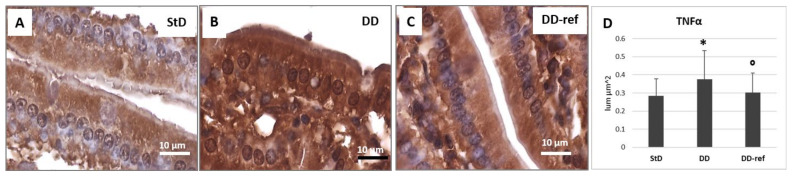
TNF-α immunostaining. Moderate staining is expressed at the baseline level inside all enterocytes in the StD-fed mice (**A**). DD strongly increases the staining (**B**), while refeeding restores it to intensity comparable to the initial values (**C**). Scale bar = 10 µm. (**D**) Graph representing the mean value of optical density according to diets. * *p* < 0.05 vs. StD, ° *p* < 0.05 vs. DD.

**Figure 12 nutrients-13-01089-f012:**
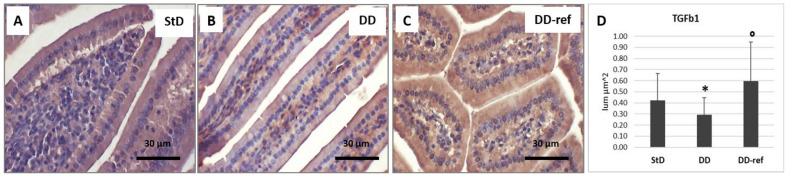
TGF-β1 immunostaining. Faint staining is expressed at the baseline level (**A**). Differently, DD strongly decreases the immune reactivity (**B**). Refeeding restored the basal intensity (**C**). Scale bar = 30 µm. (**D**) Graph representing the mean value of optical density according to diets. * *p* < 0.05 vs. StD, ° *p* < 0.05 vs. DD.

**Figure 13 nutrients-13-01089-f013:**
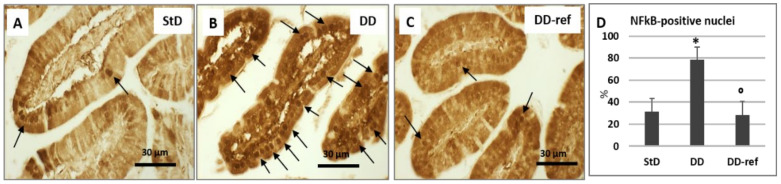
NF-κB immunostaining. At the baseline level, faint to very faint staining is expressed inside the enterocytes cytoplasm, and very few nuclei (arrows) are stained (**A**). After DD feeding, the staining increased strongly and many nuclei (arrows) are intensely stained (**B**). Refeeding restore the basal cytoplasmic staining intensity and nuclear staining (**C**). Scale bar = 30 µm. (**D**) Graph representing the % of stained nuclei according to diets. * *p* < 0.05 vs. StD, ° *p* < 0.05 vs. DD.

**Table 1 nutrients-13-01089-t001:** Diet composition. * Nitrogen (%) from free AA only. ° Nitrogen (%) from vegetable and animal proteins and added AA. StD = Standard diet; DD = non-essential amino acids rich diet. The black line represents the limit between EAAs (upside) and NEAAs (beneath). bcaa = branched chain amino-acids.

	StD	DD
KCal/Kg	3952	3995
Carbohydrates %	54.61	61.76
Lipids %	7.5	6.12
Nitrogen %	21.8 °	20 *
Proteins: % of total nitrogen content	95.93	0
Free AA: % of total nitrogen content	4.07	100
EAA/NEAA (% in grams)	-	15/85
Free AA composition (%)		
L-Leucine (bcaa)	-	4.7
L-Isoleucine (bcaa)	-	2.35
L-Valine (bcaa)	-	2.35
L-Lysine	0.97	2.44
L-Threonine	-	13.13
L-Hystidine	-	5.65
L-Phenylalanine	-	4
L-Cystine	0.39	-
L-Cysteine	-	5.65
L-Methionine	0.45	1.9
L-Tyrosine	-	9.25
L-Triptophan	0.28	0.04
L-Alanine	-	30
L-Glycine	0.88	12.7
L-Arginine	1.1	11.8
L-Proline	-	10.2
L-Glutamine	-	10.2
L-Serine	-	5.1
L-Glutamic Acid	-	2
L-Asparagine	-	1.4
L-Aspartic Acid	-	0.8

**Table 2 nutrients-13-01089-t002:** Feces production (mean ± st.dev.). * *p* < 0.05 vs. StD, ° *p* < 0.05 vs. DD.

	StD	DD	DD-Ref	*F*	*p*
Feces (g/day)	0.66 ± 0.05	0.32 ± 0.02 *	0.51 ± 0.15 *°	29.27	0.000

**Table 3 nutrients-13-01089-t003:** Body weight and organ weight (mean ± st.dev.) at the end of treatment. rpWAT, retroperitoneal white adipose tissue. BAT, brown adipose tissue. * *p* < 0.05 vs. StD, ° *p* < 0.05 vs. DD.

	StD	DD	DD-Ref	*F*	*p*
Body weight (g)	29.67 ± 1.97	17.83 ± 1.17 *	28.13 ± 1.2 °	202.84	0.000
Body length (cm)	10.08 ± 0.28	9.43 ± 0.08 *	9.59 ± 0.09 *	36.52	0.000
Heart (g)	0.18 ± 0.02	0.13 ± 0.01 *	0.14 ± 0.01 *	32.80	0.000
Kidneys (g)	0.58 ± 0.06	0.30 ± 0.04 *	0.49 ± 0.04 °	87.05	0.000
Liver (g)	1.68 ± 0.2	0.65 ± 0.07 *	1.38 ± 0.12 *°	143.03	0.000
Spleen (g)	0.15 ± 0.04	0.05 ± 0.01 *	0.08 ± 0.01 *°	44.02	0.000
rpWAT (g)	0.21 ± 0.02	0.04 ± 0.01 *	0.19 ± 0.03 °	173.8	0.000
BAT (g)	0.17 ± 0.01	0.08 ± 0.01 *	0.34 ± 0.04 *°	271.2	0.000
Triceps surae (g)	0.35 ± 0.03	0.18 ± 0.02 *	0.28 ± 0.04 *°	68.05	0.000

**Table 4 nutrients-13-01089-t004:** Blood parameters (mean ± st.dev.) at the end of treatment. * *p* < 0.05 vs. StD, ° *p* < 0.05 vs. DD.

	StD	DD	DD-Ref	*F*	*p*
Glucose (mg/dL)	136 ± 7.16	108.3 ± 6.76 *	128.2 ± 15.95 °	13.94	0.000
Erythrocytes (RBC) (M/µL)	9.7 ± 0.42	10.37 ± 0.3 *	9.31 ± 0.27 °	20.69	0.000
Hemoglobin (g/dL)	15.08 ± 0.48	15.12 ± 0.47	14.91 ± 0.9	0.24	0.791
Platelet (PLT) (K/µL)	1278.0 ± 516.4	540.1 ± 291.32 *	1115.3 ± 65.2 °	10.14	0.000
Leucocytes (WBC) (K/µL)	7.82 ± 1.76	3.76 ± 1.19 *	6.0 ± 0.93 *°	18.45	0.000
Neutrophils (K/µL)	1.78 ± 0.42	2.24 ± 0.6	1.40 ± 0.41 °	6.03	0.009
Lymphocytes (K/µL)	5.76 ± 1.37	1.50 ± 0.8 *	4.42 ± 0.49 *°	41.54	0.000
N/L ratio (%)	0.32 ± 0.1	1.83 ± 0.6 *	0.31 ± 0.07 °	48.98	0.000
**Serum**					
Albumin (g/L)	28.66 ± 2.57	25.43 ± 1.64	28.42 ± 3.2	3.98	0.034
Urea (mmol/L)	8.54 ± 2.01	8.78 ± 1.17	7.58 ± 0.72	1.63	0.219
Creatinine (µmol/L)	23.83 ± 1.89	35.63 ± 4.4 *	28.0 ± 4.34 °	20.58	0.000
Aptoglobin (mg/mL)	2.70 ± 1.05	0.14 ± 0.01 *	2.80 ± 1.02 °	25.46	0.000

**Table 5 nutrients-13-01089-t005:** Urine parameters (mean ± st.dev.) at the end of treatment. * *p* < 0.05 vs. StD, ° *p* < 0.05 vs. DD.

	StD	DD	DD-ref	*F*	*p*
Albumin (g/L)	2.66 ± 0.23	1.36 ± 0.4 *	2.13 ± 0.7 °	18.3	0.000
Urea (mmol/L)	835.9 ± 35.85	1445.2 ± 181.7 *	1396.4 ± 231.3 *	19.59	0.000
Creatinine (µm/L)	6147.66 ± 957.8	4637.8 ± 396.5 *	4414 ± 240.23 *	11.78	0.000
Total protein (g/L)	2.29 ± 0.2	1.38 ± 0.11 *	2.48 ± 0.52 °	16.08	0.000

**Table 6 nutrients-13-01089-t006:** Descriptive statistics of morphometric variables of the jejunum of mice fed with StD and DD and after refeeding with StD (DD-ref). * *p* < 0.05 vs. StD, ° *p* < 0.05 vs. DD.

	StD	DD	DD-Ref	*F*	*p*
Wall thickness (µm)	346.2 ± 41.6	319.9 ± 32.7	341.1 ± 32.7	3.02	0.057
Muscular layer thickness (µm)	68.6 ± 8.4	60.0 ± 6.9 *	66.8 ± 6.93 °	7.43	0.001
Mucosa layer thickness (µm)	277.56 ± 44	259.94 ± 37.2	274.31 ± 33.2	1.19	0.311
Villi length (µm)	208.86 ± 38.3	187.21 ± 38.6	194.31 ± 33.1	1.83	0.169
Villi diameter (µm)	73.1 ± 11.8	73.6 ± 12.6	74.7 ± 10.2	0.11	0.898
Epithelium thickness (µm)	28.5 ± 6.5	20.7 ± 4.7 *	26.3 ± 3.5 °	12.94	0.000
Enterocytes (*n°* Nuclei/100 µm)	18.5 ± 5.1	11.4 ± 2.6 *	14.6 ± 2.1 *°	18.2	0.000
Enterocytes: Nuclear diameter (µm)	4.6 ± 0.8	5.2 ± 0.9	4.5 ± 0.88 °	4.46	0.016
Enterocytes: junctions (*n°*/100 µm)	24.36 ± 3.4	16.88 ± 3.5 *	21.41 ± 4.3 °	23.42	0.000
Goblet cells (nr/mm)	37.4 ± 13.4	23.0 ± 9.2 *	31.9 ± 9.1 °	9.13	0.000
Crypt depth (µm)	76.93 ± 7.3	70.38 ± 8.0	79.99 ± 8.1	0.88	0.422
Paneth cells granules (nr/cell)	23.12 ± 5.6	17.94 ± 3.2 *	21.18 ± 5.1	5.57	0.007

## Data Availability

The data presented in this study are available on request from the corresponding author.

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
