# Peer review of "Qualitative Nitrogen Malnutrition Damages Gut and Alters Microbiome in Adult Mice. A Preliminary Histopathological Study"

_nutrients, 2021, doi:10.3390/nu13041089_

Round 1

Reviewer 1 Report

In this manuscript, the authors described the effect of dietary unbalanced EAA/non-EAAs ratio on gut homeostasis and microbiome. They reported that the excess non-EAA could cause damage to the epithelial tight junction. Besides, they found that excess non-EAA increased the probiotics and decreased the pathogens in the gut. It is an interesting study by discovering qualitative nitrogen malnutrition in impairing the gut epithelium and altering the microbiota composition. 

Comments

  1. The introduction should focus on the relationship between malnutrition, gut disease, and gut microbiome based on the title of the manuscript. 
  2. There were no H&E-stained images with high magnification to compare epithelial integrity and inflammation status among different groups. 
  3. The gut microbiota results have unclear significance as the excreted faces rather than the disease-relevant jejunum luminal contents were analyzed. Moreover, it’s not clear why microbiota analysis is so limited to those species. Should add more discussions and double-check how to present microbiota in the text, for example, italic.
  4. The resolution of all images is pretty low. It’s hard to tell the difference among different groups. Different sizes and types of fonts were used in the figures. The scale bars in the images are inconsistent with those described in the figure legends, such as figs. 5, 10, and 11.
  5. The manuscript is poorly prepared. The text and figs have many errors and typos, such as “gender” in line 87, “f.c.u” in line 252, etc. The discussion contains too much-unrelated information. 

Author Response

we thank this referee for the precious suggestions that we gladly welcome.

1. We introduced the concept of malnutrition in order to better understand the link with the intestinal flora described in the other sections of the introduction.

2. We added figure 4 in order to highlight the differences of the intestinal epithelium with diets. We used semi-fine resin sections (1micron thickness) in order to better highlight the morphological details. Additionally, we added Figure 8 as a representative of the DD diet-induced inflammatory state.

3. We thank and agree with the referee. The feces were collected immediately after euthanasia directly inside the intestine. The choice of the number of species analyzed was dictated by a practical consideration. Indeed, we did not want to provide a “fecal fingerprint”. Our aim was to identify saprophytes and minor intestinal pathological and Mycétes species able to activate immunological responses using a simple, universally available and cheap technique on easily available intestinal contents (Pasini et al., 2016, doi: 10.1016/j.jchf.2015.10.009; Pasini et al., 2019, doi: 10.23736/S0026-4806.18.05589-1). The immunomodulation of the intestinal wall is strongly influenced by the entire microbial population except in acute pathologies (gastroenteritis). The hosts of the intestine are classified according to the stimulus exerted on the immune system, this is why we have selected Lactobacillus spp, Escherichia coli, total coliforms and Mycétes. Indeed, Lactobacillus spp represent the microorganisms with greater protective and immunomodulating action (O'Callaghan et al., 2013, doi: 10.1007/82_2011_187; Pourramezan et al., 2018, doi: 10.1007/s12223-017-0531-x). Malnutrition can reduce this population by causing the overgrowth of other commensal species (Lin et al., 2017, doi: 10.1007/s00726-017-2493-3; Ma et al., Protein Pept Sci. 2017;18(8):795-808). The reduction in the production of bacteriocins by Lactobacillus spp allows the overgrowth of Escherichia Coli, coliforms and Mycétes generates an inflammatory response (Zeng et al., 2017, doi: 10.1038/mi.2016.75; Li et al., 2014; Ott et al., 2018). The metabolites produced by the altered gut flora modify the adaptive responses of the intestinal mucosa. Probably not only these groups of bacteria generate the inflammatory mechanism described. So we cannot rule out the involvement of other bacterial families which would be interesting to investigate in the future.

In the text we had presented the microbiota in italic.

4. We are sorry. We have corrected the errors indicated. Unfortunately the size of figures is limited by the available space. So even the images are highly magnified (100x oil, like figures 4, 5 and 7) are penalized.

5. We are sorry. We have corrected the errors indicated and re-checked all text.

We thank and agree with the referee that discussion contains a lot of information regarding very different topics. This can probably make the text feel non-linear and with unrelated information. However, this apparent non-linearity we believe has allowed us to (hopefully) discuss the individual data more clearly in order to provide the reader with a general logical framework that justifies our conclusions.

Reviewer 2 Report

The article discussed about vital food components and its effect on intestinal health. Dietary components like fiber, fat changes microbiota and intestinal barrier function. This research revealed that Amino acids composition of the diet changes microbiota and its effect barrier function, intestinal epithelial cell functions and inflammatory molecules. The research is well designed and  executed and discussed about intestinal morphology. 

I have few questions,

  1. With reduced PAS staining and Paneth cells granules, DD group mice had any increase in CFU no of gram (+) and gram (-) bacteria in agar culture.
  2. With increased inflammatory markers in the intestinal tissue after DD supplemented group.  Author's included in the immune cell infiltration picture and scoring increases readers interest.
  3. With decreased body weight in the DD group, did any difference in the lipid profile. Because, previous reports indicates amino acids supplementation decreases body weight gain and improves fatty liver disease on high fat diet.

Author Response

we thank this referee for the precious suggestions that we gladly welcome.

1. Gram staining highlights the morphological / structural characteristics of the bacteria that do not necessarily coincide with their metabolic and pathogenic characteristics. The reduction of PAS staining and cytoplasmic granules of Paneth cells in mice fed the DD diet corresponds to a variation of bacterial flora whose composition cannot simply be referred to the Gram stain. In fact, what we have observed is that in the animals of the DD group, the Lactobacillus spp (which are Gram -), decrease, while the Coliforms (which are gram +), increase. It should also be emphasized that both Lactobacillus and E. Coli are Gram - although they perform different and opposing functions in the intestine.

2. We added Figure 8 as a representative of the DD diet-induced inflammatory state. In materials and methods section we have included the reference to the scoring scale used to evaluate the inflammatory state.

3. Thank you for the question which represents an interesting topic on which we are working. In this work we did not do the lipid profile, however the data that emerged is the significant reduction in both fat mass (WAT and BAT) and muscle mass (Table 3). We consider this result equally interesting as a qualitatively poor diet acts simultaneously and drastically on the two tissues providing a possible basis for the metabolic impairment of the whole organism.

Round 2

Reviewer 1 Report

All of the queries raised in my report have been considered in full detail and the revisions in the new manuscript address these completely. There are no further issues to address. I would like to thank the authors for their detailed and professional response, which made it a pleasure to review their work.